# OSLO:
# One-Shot Label-Only Membership Inference Attacks

**Yuefeng Peng**
University of Massachusetts Amherst
yuefengpeng@cs.umass.edu

**Jaechul Roh**
University of Massachusetts Amherst
jroh@umass.edu

**Subhransu Maji**
University of Massachusetts Amherst
smaji@cs.umass.edu

**Amir Houmansadr**
University of Massachusetts Amherst
amir@cs.umass.edu

## Abstract

We introduce One-Shot Label-Only (OSLO) membership inference attacks (MIAs), which accurately infer a given sample's membership in a target model's training set with high precision using just *a single query*, where the target model only returns the predicted hard label. This is in contrast to state-of-the-art label-only attacks which require $\sim 6000$ queries, yet get attack precisions lower than OSLO's. OSLO leverages transfer-based black-box adversarial attacks. The core idea is that a member sample exhibits more resistance to adversarial perturbations than a non-member. We compare OSLO against state-of-the-art label-only attacks and demonstrate that, despite requiring only one query, our method significantly outperforms previous attacks in terms of precision and true positive rate (TPR) under the same false positive rates (FPR). For example, compared to previous label-only MIAs, OSLO achieves a TPR that is at least $7\times$ higher under a 1% FPR and at least $22\times$ higher under a 0.1% FPR on CIFAR100 for a ResNet18 model. We evaluated multiple defense mechanisms against OSLO.

## 1 Introduction

Deep learning (DL) models are vulnerable to membership inference attacks (MIAs), where an attacker attempts to infer whether a given sample is in the target model's training set [1]. The success of such MIAs may lead to severe individual privacy breaches as DL models are often trained on private data such as medical records [2] and facial images [3]. Existing black-box MIAs fall into two categories: *score-based* MIAs [1, 4, 5] and *decision-based* MIAs [6, 7], also known as *label-only* MIAs. Score-based attacks assume access to the model's output confidence scores, which can be defended against if the model only outputs a label. In contrast, label-only attacks infer membership based on the labels returned by the target model, which is a stricter and more practical threat model.

State-of-the-art label-only MIAs measure the sample's robustness to adversarial perturbations, classifying samples with higher robustness as members. This robustness serves as a proxy for model confidence, which is typically higher for training samples. These attacks utilize *query-based* black-box adversarial attacks [8, 9] to determine the amount of necessary adversarial perturbation for each sample. However, accurately estimating this requires a large number of queries to the model, sometimes up to thousands for each sample. This approach is not only costly but also easily detectable [10]. Furthermore, these attacks often lack precision and are unsuccessful in the low false positive rate regime, as indicated by previous studies [11] and our evaluations. In summary, existing label-only MIAs are not practical.

38th Conference on Neural Information Processing Systems (NeurIPS 2024).

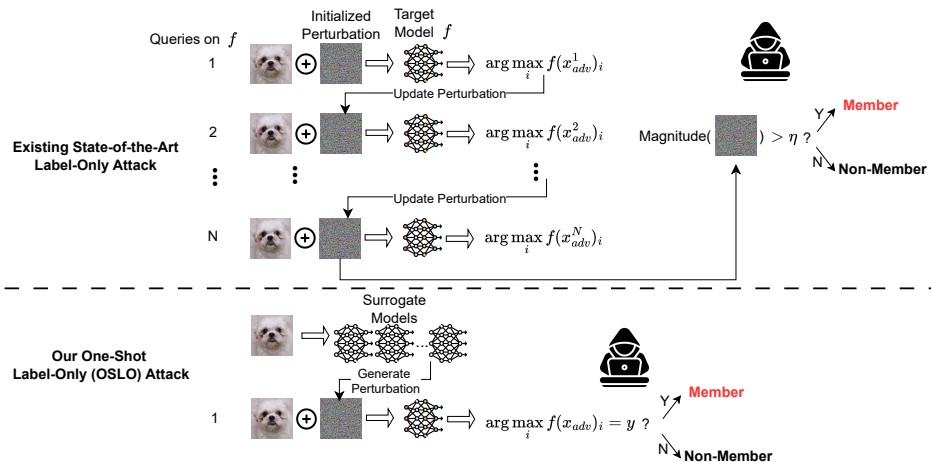

Figure 1: **An illustration of OSLO versus the state-of-the-art boundary attack.** Boundary attack requires querying the target model thousands of times, whereas OSLO requires only a single query.

Table 1: **Comparison to previous label-only attacks.** We report the highest attack precision that these attacks can achieve with a recall greater than 1% on CIFAR-10 using a ResNet18 model.

| Attack method | Require no knowledge about target model architecture | Require no auxiliary data | Attack precision | Num of queries used |
|---|---|---|---|---|
| Transfer Attack [6] | ✗ | ✗ | 60.9% | $|\mathcal{D}_{shadow}|$ |
| Data Augmentation [7] | ✓ | ✓ | 69.2% | $\sim 10$ |
| Boundary Attack (Gaussian Noise) [7] | ✓ | ✓ | 61.3% | $\sim 700$ |
| Boundary Attack (HopSkipJump) [7, 6] | ✓ | ✓ | 64.5% | $\sim 6000$ |
| Boundary Attack (QEBA) [6] | ✓ | ✓ | 71.4% | $\sim 6000$ |
| OSLO (Ours) | ✓ | ✗ | **96.2%** | 1 |

We propose novel One-Shot Label-Only (OSLO) MIAs that can infer a sample's membership in the target model's training set with just *a single query*, even when the target model only returns hard labels for input samples. Table 1 compares OSLO to previous label-only MIAs. OSLO is based on constructing transfer-based adversarial example, with just enough magnitude of adversarial perturbation added to each sample to cause misclassification if the sample is non-member (but not enough if it is a member) based on a set of source and validation models, a technique that was initially proposed in [12]. Prior work has considered using a fixed threshold of adversarial perturbation across all samples to infer membership status, resulting in a lower precision [7, 6].

We extensively compared OSLO with state-of-the-art label-only methods [7, 6]. Following recent standards, we evaluated the attacks using metrics such as the true positive rate (TPR) under a low false positive rate (FPR) and performed a precision/recall analysis. Our results show that previous label-only MIAs perfom poorly, exhibiting high false positive rates. In contrast, OSLO achieves high precision in identifying members, outperforming prior work by a significant margin. For example, as shown in Figure 2, OSLO is 5× to 67× more powerful than other label-only MIAs in terms of TPR under 0.1% FPR across three datasets using a ResNet18 [13] model. OSLO achieves over 95% precision on all three datasets on ResNet18 model with a recall greater than 0.01, while the highest precision achieved by other attacks is only 71.4%, 84.2%, and 67.9% respectively. Furthermore, we conduct a detailed comparison of OSLO with previous work, exploring why previous label-only attacks fail to achieve high precision.

## 2 Background and preliminaries

### 2.1 Adversarial attacks

Early works [14, 15] have demonstrated the susceptibility of deep neural networks (DNNs) to adversarial attacks. These attacks involve crafting subtle perturbations to input data, causing a DNN to

make incorrect predictions while the perturbations remain almost imperceptible to humans. Adversarial attacks can significantly compromise model performance, resulting in issues like misclassification or misgeneration. These attacks are typically classified based on the adversary's level of access to the model [16]: white-box and black-box settings. In white-box attacks [17, 15, 18, 19, 20, 21], adversaries have full knowledge of the victim model, including its architecture, parameters, and gradients. In contrast, black-box attacks [22, 23] limit adversaries to querying the model and obtaining output labels or confidence scores. We primarily discuss black-box attacks to align with the threat model of our MIA settings.

**Black-box adversarial attacks.** Black-box attacks primarily include two methods: *(i)* **Query-based** attacks iteratively query the target model to gather information and craft adversarial examples, adjusting perturbations based on the model's feedback. These attacks often require many queries to be effective. *(ii)* **Transfer-based** attacks generate adversarial examples using a white-box model and then transfer these examples to attack another model. Previous works have proposed various transfer-based attacks. For instance, Momentum Iterative Fast Gradient Sign Method (MI-FGSM) [21] introduces momentum iterative gradient-based methods for generating adversarial examples to effectively attack both white-box and black-box models. Diverse Inputs Iterative Fast Gradient Sign Method (DI$^2$-FGSM) [24] enhances the transferability of adversarial examples by applying random transformations to input images at each iteration of the attack process. Translation-Invariant Fast Gradient Sign Method (TI-FGSM) [25] optimizes perturbations over a set of translated images to make the adversarial examples less sensitive to specific model features and improve their transferability. Admix [26] enhances traditional input transformations by mixing the input image with images from other categories to create admixed images. These admixed images are used to compute gradients, helping to create more transferable adversarial examples.

## 2.2 Membership inference attacks

Membership inference attacks (MIAs) [27, 28, 29, 30, 31, 32, 33] aim to determine whether a specific data sample is a member of the training dataset of a target model. Existing MIAs often assume that the attacker can access the target model's confidence scores to infer membership. These score-based attacks exploit the difference in confidence scores for training versus non-training data, as models often exhibit higher confidence for samples they have seen during training [1, 4, 5]. Such attacks can be easily defended by limiting access to only the predicted labels, which reduces the amount of information available to the attacker.

**Label-Only MIAs.** Compared to score-based attacks, a more practical and stringent type of attack is decision-based or label-only MIAs [7, 31], which operate under the constraint of having access only to the final decision labels. Several label-only attacks have been proposed. For example, inspired by the transferability of adversarial examples, transfer attacks [31] explore the transferability of membership information. They use a shadow dataset labeled by the target model to train a shadow model and perform a score-based attack on the shadow model, expecting that the differences between members and non-members in the target model will reflect similarly in the shadow model. Data augmentation attacks [7] exploit the robustness of samples to data augmentation transformations to differentiate members from non-members, with higher robustness indicating membership. Similarly, state-of-the-art label-only MIAs, known as boundary attacks [31, 7], use the robustness of samples to strategic input perturbations as label-only proxies for the model's confidence, where higher robustness implies a higher confidence score. Boundary attacks have different methods for adding perturbations. For example, Gaussian noise is one method where the attacker continuously adds Gaussian noise until the label changes. Attackers also use adversarial attacks to measure the robustness of samples to adversarial perturbations. Specifically, query-based adversarial attacks like HopSkipJump [9] and QEBA [8] algorithms are used to add perturbations to each sample to generate adversarial examples.

Existing label-only MIAs typically require many queries to be effective [6, 7] and struggle to achieve high TPRs at low FPRs [11]. A recent work YOQO [34] reduced the query budget. However, by design, it cannot adjust the TPR-FPR trade-off and remains ineffective in the low FPR regime, limiting its practicality.

# 3 OSLO

## 3.1 Problem definition

Following Ye et al. [35] and Leemann et al. [36], we view MIA as a hypothesis testing problem:

$$H_0 : (x, y) \notin D \quad vs \quad H_1 : (x, y) \in D \tag{1}$$

Successfully rejecting the null hypothesis equates to determining that the sample $(x, y)$ is a member in the target model's training set $D$. Given a sample for inference, we initially assume that the sample is a non-member ($H_0$). Our objective is to design an attack method that can reject the null hypothesis with high confidence.

## 3.2 Threat model

**Attacker's capability.** Given a target model $f$, we assume that the attacker can only access $f$ in a black-box manner, meaning that means that for any input $x$, the attacker can only observe $f(x)$ but cannot access the internal parameters $\theta$ or any intermediate outputs of $f$. Furthermore, we consider a more restrictive scenario: label-only access, where the attacker can only obtain the predicted labels (i.e., $\hat{y} = \arg\max f(x)$) from the target model, not the confidence scores or probabilities associated with each class. This label-only, black-box attack scenario represents one of the most challenging and realistic settings for conducting attacks against DL models. Similar to previous attacks [1, 7], it is assumed that the attacker possesses an auxiliary dataset $D_{\text{aux}}$ that shares the same distribution as the target model's training set for the purpose of training their surrogate models. However, we do not assume the attacker has any knowledge of the target model, including its architecture.

**Attacker's goal.** The attacker's objective is to reliably infer whether a given sample is a member of the target model's training set with high precision. As underscored by Carlini et al. [5], identifying individuals within sensitive datasets with high precision poses a significant privacy risk, even if only a few users are correctly identified. Moreover, the precision of these attacks is not just a concern for individual privacy; it also lays the groundwork for more advanced extraction attacks [37], making the pursuit of high precision in MIAs a critical focus for attackers. Additionally, the attacker aims to infer membership with as few queries as possible, as excessive querying not only incurs significant costs but is also detectable by defenders [10]. Our OSLO limits the query budget to a single query.

## 3.3 Intuition of OSLO

Previous work [4, 1] has demonstrated that target models tend to be overconfident on members. existing label-only attacks [6, 7] suggest that models are more confident about members, making it more diffficult to "change their mind" on these samples. Their experiments also show that members are usually more robust to adversarial perturbations than non-members. We believe that the differences in the required adversarial perturbation between members and non-members can manifest in two ways: (i) on average, members tend to require more adversarial perturbation than non-members; and (ii) for each individual sample, the member status may demand more adversarial perturbation than the non-member status. Previous label-only MIAs [6, 7] have exploited the former observation, but this has led to high false positive rates and low precision. Our OSLO is based on the latter. We add just enough adversarial perturbation to cause a misclassification for the sample if it is a non-member but not if it is a member. If the sample is not successfully misclassified, it indicates that the added perturbation was insufficient, and the sample is likely a member. Note that here, 'sufficient perturbation' refers to a variable standard for each sample, changing with the sample, rather than a uniform standard for all samples.

## 3.4 Attack method

Previous label-only MIAs face two main issues: *low precision* and *high query budget*. We address both issues simultaneously by proposing our *transfer-based* adversarial attack based attack, which implements the aforementioned hypothesis testing framework. Specifically, we first train a group of surrogate models. For each sample, we construct a transferable adversarial example using the surrogate model(s). Then, we input the crafted adversarial example into the target model and obtain

the predicted label. If the sample is misclassified, it indicates that the adversarial example has successfully transferred, implying that there is not enough evidence to reject the null hypothesis $H_0$. Otherwise, we reject $H_0$ and determine the sample as a member. It is worth noting that the perturbation added to each sample should be sufficient for it to transfer as a non-member but not as a member. Therefore, carefully controlling the scale of perturbation added to each sample is required. The framework of OSLO is divided into three stages: *surrogate model training*, *transferable adversarial example construction*, and *membership inference*.

**Surrogate model training.** The attackers first train their own surrogate models to construct adversarial examples. As assumed in Section 3.2, we assume that the attacker has an auxiliary dataset with the same distribution as the target model's training set. The attacker trains a group of models with different structures on this dataset, training $N$ models for each structure. These models are then divided into source models and validation models based on different purposes in the adversarial example generation stage. Source models are used to calculate gradients for generating adversarial perturbations, while validation models are used to control the magnitude of the added perturbations.

**Transferable adversarial example generation.** After the source models are trained, the attacker generates adversarial examples on these models using transfer-based adversarial attacks. Existing transfer-based attacks typically search for adversarial examples within a fixed $l_p$-norm ball and do not minimize the perturbation added. In other words, these attacks allocate a uniform perturbation budget $\epsilon$ to all samples, accepting any adversarial examples found within this budget. While this approach is reasonable in the context of general adversarial attacks, it is not suitable for our MIA framework, as OSLO requires finer-grained control over the amount of perturbation added.

In OSLO, the perturbation budget for each sample is adaptive. Specifically, we utilize the Geometry-Aware (GA) framework proposed in [12], which employs a set of validation models to regulate the amount of perturbation added to each sample. The process of adding perturbations is incremental. During this process, validation models are used to monitor the transferability of the sample. The perturbation addition process is terminated early—referred to as early stopping—when the perturbation added is sufficient for the sample to deceive the validation model, and the confidence of the correct class on the validation model drops below a specified threshold $\tau$:

$$\mathbb{P}(\hat{h}(x) = y) = \frac{\exp(h_y(x; \theta))}{\sum_j \exp(h_j(x; \theta))} < \tau. \quad (2)$$

---

**Algorithm 1:** Transferable adversarial example generation.

---

**Input:** Benign input $x$ with label $y$; source models $g$; validation model $h$; number of sub-procedures $K$; number of iterations $N$; step size $\alpha$; maximum perturbation size $\epsilon$ and threshold $\tau$;

**Output:** Transfer-based unrestricted adversarial example $x'$ with approximately minimum change;

$x_0 \leftarrow x$;
**for** $k = 1, 2, \ldots, K$ **do**
    $x'_0 \leftarrow x_{k-1}$;
    **for** $i = 1$ **to** $N$ **do**
        Compute gradient:
        $g_i \leftarrow \nabla_x L(g(x'_{i-1}, y)$;
        Update adversarial example using the update function:
        $x'_i \leftarrow \text{Update}(x'_{i-1}, g_i, \alpha)$;
        Clip $x'_i$ to ensure it is within the $\frac{k\epsilon}{K}$-ball of $x$:
        $x'_i \leftarrow \text{clip}(x'_i, x - \frac{k\epsilon}{K}, x + \frac{k\epsilon}{K})$;
        $\text{conf} \leftarrow \frac{\exp(h_y(x'_i))}{\sum_j \exp(h_j(x'_i))}$;
        **if** *conf* $< \tau$ **then**
            **return** $x'_i$; // early-stopping criterion
        **end**
    **end**
    $x_k \leftarrow x'_N$;
**end**
**return** $x_K$;

---

where $h_j(x; \theta)$ denotes the logits for class $j$ produced by the validation model, and $y$ is the ground truth class. Through this method, the process can be terminated when the transferability of each sample reaches a certain level, resulting in tailored perturbations for each sample. This can more effectively distinguish the differences of the sample in the *in* and *out* worlds.

Algorithm 1 outlines our method for generating transferable adversarial examples, adapted from Liu et al.'s GA framework [12] to align with our objective. The algorithm inputs with a benign sample $x$ with its corresponding label $y$, a set of source models denoted as $g$, and a set of validation models $h$. The algorithm proceeds through $K$ sub-procedures, each conducting an iterative refinement of the

adversarial example across $N$ iterations, guided by a pre-established step size $\alpha$ and bounded by a maximal perturbation magnitude $\epsilon$. At each iteration, the gradient of the loss function $L$ with respect to the adversarial input is computed, informing the perturbative update. This process is modulated by the validation model $h$. If at any step the confidence of the ground-truth class falls below the threshold $\tau$, the early stopping is triggered to prevent over-perturbation, thus ensuring that the perturbations are precisely calibrated for each sample.

**Membership inference.** Finally, for each sample $x$, the attacker inputs the transferable adversarial example $x'$, generated by Algorithm 1, into the target model $f$. The classification outcome determines the evidence against the null hypothesis $H_0$. If $\arg\max_i f(x')_i \neq y$, it suggests insufficient evidence to reject $H_0$. In contrast, a classification of $\arg\max_i f(x')_i = y$ allows us to reject $H_0$, hence classifying $x$ as a member. Our membership inference strategy seeks to reject $H_0$ with high confidence, aiming to achieve high precision in the identification of members.

## 4 Evaluation

### 4.1 Experimental setup

**Datasets.** We utilized three datasets commonly used in prior works [5]: CIFAR-10 [38], CIFAR-100 [38]and SVHN [39]. For CIFAR-10 and CIFAR-100, we selected 25,000 samples to train the target model under attack. For SVHN, we randomly selected 2,000 samples to train the target model. For each dataset, we trained the surrogate models used for the attack with a set of data that is the same size as the training set of the target model but disjoint. Further details on dataset splits can be found in Appendix A.1.1.

**Models.** We adopted the widely used ResNet18 [13] and DenseNet121 [40] as the target model architectures. In addition to these two, we incorporated five additional model architectures as surrogate models. Detailed training and attack setup are provided in Appendix A.1.

**Baseline Attacks.** We compare OSLO against five state-of-the-art label-only MIAs, including the transfer attack [6], three boundary attacks [7, 6], and the data augmentation attack [7]. Detailed settings are provided in Appendix A.1.4. We exclude YOQO [34] from our evaluation, as it is not compatible with our evaluation metrics (see Appendix A.6).

**Metrics.** We employ the following metrics to evaluate the effectiveness of the attacks: *(i)* attack **TPR and FPR** [5]: Carlini et al. suggest that membership inference attacks should be analyzed using TPR and FPR, especially considering that TPR in a low FPR setting is crucial for assessing the success of MIAs. We report the log ROC curve to show the TPR and FPR of the attack, with a focus on the low FPR regime; *(ii)* attack **precision and recall**: Attack precision reflects how reliably an attack can infer membership. High-precision attacks allow an attacker to credibly violate the privacy of a portion of the samples. Specifically, precision is defined as the proportion of true members correctly identified among all positively predicted members by an adversary. Meanwhile, recall quantifies the proportion of true members that the adversary has correctly identified out of all actual members. We discarded average-case metrics such as accuracy and AUC, as they have been shown not to accurately reflect the threat of MIAs [5].

### 4.2 Results

#### 4.2.1 Evaluating attack TPR under Low FPR

We first evaluate six attacks, including OSLO, focusing on TPR under low FPR regime. For OSLO, we control the trade-off between TPR and FPR by adjusting the threshold $\tau$ during the transferable adversarial example generation phase. A lower $\tau$ results in more perturbation added to each sample. Consequently, if the samples still fail to deceive the target model, it is more likely that they are members, thereby reducing FPR. For other attacks, we adjust their corresponding parameters to manage TPR and FPR. For instance, in the boundary attack, increasing the perturbation threshold for identifying members ensures that only samples requiring more significant perturbations are classified as members. Based on the assumption that members generally need greater perturbation than non-members, this method should minimize the number of samples mistakenly identified as members and enhance the credibility of the attack.

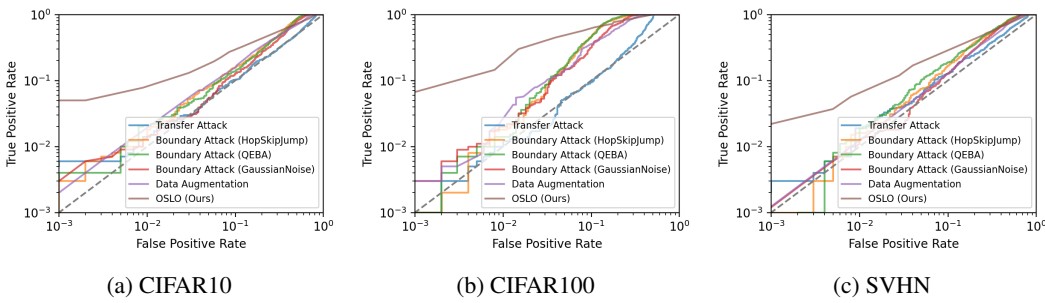

(a) CIFAR10         (b) CIFAR100         (c) SVHN

Figure 2: **ROC curves for various label-only attacks on three different datasets on ResNet18.** Each line represents the TPR of an attack under different FPRs, with an emphasis on the low-FPR regime using a logarithmic scale.

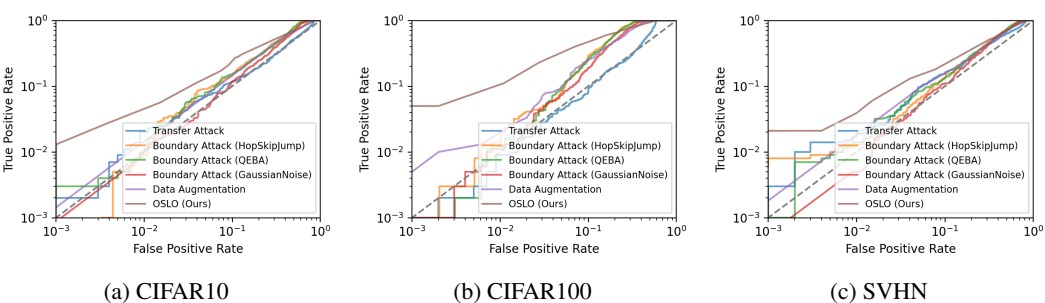

(a) CIFAR10         (b) CIFAR100         (c) SVHN

Figure 3: **ROC curves for various label-only attacks on three different datasets on DenseNet121.** Each line represents the TPR of an attack under different FPRs, with an emphasis on the low-FPR regime using a logarithmic scale.

As shown in Figure 2, **OSLO outperforms all previous label-only MIAs by a large margin**. For example, compared to previous attacks, OSLO achieves a TPR that ranges from $5\times$ to $67\times$ higher under 0.1% FPR, and from $3\times$ to $16\times$ higher under 1% FPR on ResNet18, evaluated across three datasets. On DenseNet121, the improvements range from $2\times$ to $12\times$ under 1% FPR. It can be observed that previous attacks were unsuccessful in the low FPR regime, whereas OSLO can successfully identify some members even at low FPR. For example, on CIFAR-100 using ResNet18, at a 0.1% FPR, the best of the other label-only MIAs achieves only a 0.3% TPR, while OSLO reaches 6.7%. Under a 1% FPR, OSLO achieves a TPR of 18.9%, with other attacks reaching at most 2.7%. These comparisons show that, **although OSLO requires only one query to the target model, it can reliably identify part of the members, while previous attacks, despite needing to perform up to thousands of queries, are almost unable to credibly identify members.**

### 4.2.2 Precision and recall analysis

We then measured the trade-off between attack precision and recall. In MIAs, precision is considered a crucial metric, whereas recall is relatively less important [5]. For instance, a precision of 1.0 means that the attacker can be 100% certain that the identified samples are members, leading to significant privacy breaches, even if only a few samples are affected. Conversely, a recall of 1.0 can be achieved by classifying all samples as members, but this does not cause any privacy breach. For MIAs, a desired outcome would be the ability to trade recall for increased precision.

As shown in Figure 4 and Figure 5, **OSLO is the only one that effectively trades recall for precision.** For example, by trading recall, OSLO achieves over 90% precision across all three datasets with ResNet18 as the target model. Specifically, on CIFAR-10, CIFAR-100, and SVHN, with recall rates of 5%, 44.7%, and 9% respectively, OSLO attained precisions of 96.2%, 92.0%, and 90.9%. In contrast, the highest precision achieved by other attacks at similar recall levels was 67.4%, 82.5%, and 65.2% respectively. This demonstrates that **OSLO can identify members with high precision, a capability that all previous label-only attacks could not match.** Previous methods could not

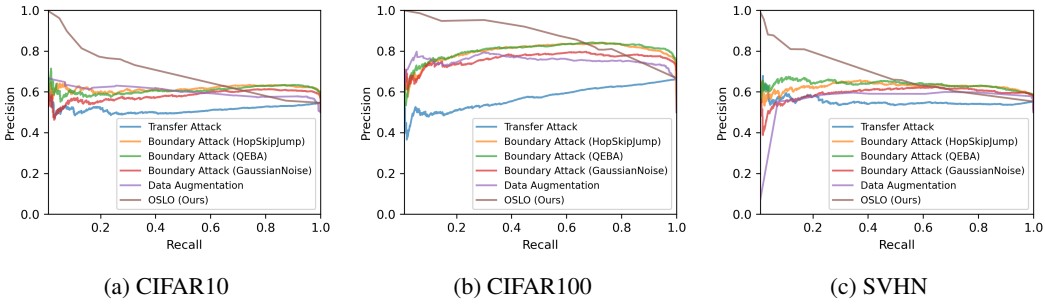

| (a) CIFAR10 | (b) CIFAR100 | (c) SVHN |

Figure 4: **Precision-Recall curves for various label-only attacks on ResNet18.** Each line represents the trade-off between precision and recall for an attack as the attack parameter is varied.

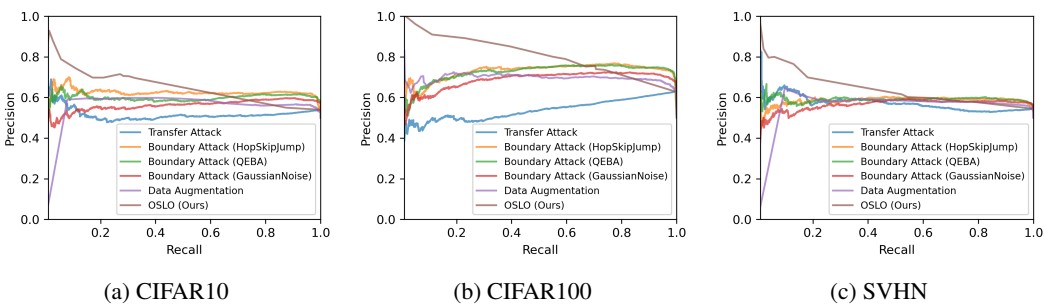

| (a) CIFAR10 | (b) CIFAR100 | (c) SVHN |

Figure 5: **Precision-Recall curves for various label-only attacks on DenseNet121.** Each line represents the trade-off between precision and recall for an attack as the attack parameter is varied.

improve precision by adjusting hyper-parameters, due to limitations inherent in their designs. We discuss this further in Section 6.1.

# 5 Ablation study

## 5.1 Effect of using different adversarial attacks

As introduced in Section 3.4, OSLO leverages transfer-based adversarial attacks. We evaluated the membership inference performance of OSLO using six different transfer-based adversarial techniques within its framework, specifically TI [25], DI [24], MI [21], Admix [26], and the combinable methods TDMI and TMDAI. The results are presented in Figure 6. Observations reveal no significant differences in the effectiveness of using these various transfer-based attacks. This may suggest that the efficacy of the attack predominantly relies on the inherent design of the OSLO framework rather than the specific choice of the individual transfer techniques.

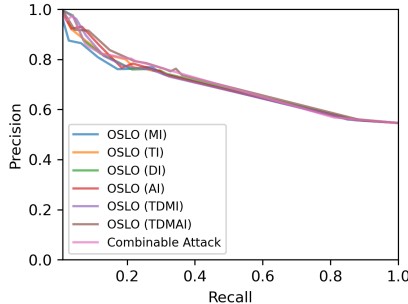

Figure 6: **Precision-Recall curve showing the attack performance of OSLO using different transfer-based adversarial attacks on CIFAR-10.**

## 5.2 Effectiveness of OSLO when target and surrogate models use different training algorithms

We assume that the attacker is unaware of the target model's architecture, using surrogate models with different structures. We further explore scenarios where the surrogate and target models are trained with different optimization algorithms. In particular, we evaluated OSLO on CIFAR-10 using ResNet18, where the target models were trained with SGD, while the surrogate models used Adam. The results are shown in Table 2.

Table 2: **OSLO performance on CIFAR-10 with ResNet18 when target and surrogate models are trained using different algorithms.**

| Target model | Source/validation models | Attack precision (Recall > 1%) | Attack TPR@1% FPR |
|---|---|---|---|
| SGD, lr=0.01 | Adam, lr=0.001 | 95.9% | 10.0% |
| SGD, lr=0.1 | Adam, lr=0.001 | 95.3% | 9.2% |

Notably, OSLO achieves over 95% precision, with a TPR exceeding 9% at 1% FPR in both settings. These results confirm that OSLO remains effective even when the target models are trained with algorithms different from those of the surrogate models.

### 5.3 Impact of validation models in OSLO

OSLO uses validation models to adjust the scale of perturbations added to adversarial examples. To understand the impact of validation models, we conducted an ablation study on CIFAR-10 using ResNet18 without the validation models, where a uniform perturbation budget was applied to all samples. The results are shown in Table 3.

Table 3: **Attack TPR and FPR of OSLO without validation models on CIFAR-10 using ResNet18.**

| Perturbation budget $\epsilon$ | TPR | FPR |
|---|---|---|
| 8/255 | 20.1% | 13% |
| 16/255 | 2.3% | 1.4% |
| 32/255 | 0.1% | 0.1% |

These results indicate that without the validation model, applying a uniform perturbation budget across all samples leads to significantly reduced attack effectiveness. For example, at $\epsilon = 32/255$, nearly all members and non-members were misclassified, resulting in both TPR and FPR being very low. This highlights the critical role of validation models in effectively calibrating perturbations.

### 5.4 Effect of using more than one shot

OSLO employs a single query to the target model to determine a sample's membership status. We also explore the potential advantages of extending OSLO to utilize multiple queries per sample, referred to as multi-shot. Specifically, Figure 7 illustrates the precision and recall of OSLO under different adversarial example generation thresholds, $\tau$.

We further report on combining results from the current threshold with those from previous, higher thresholds. For example, 'three shots' represents feeding the current shot and the previous two shots' generated images to the target model, identifying samples as members if all three adversarial examples fail to fool the target model. As shown in Figure 7, we find that using more shots does not yield a significant improvement. We attribute this to the observation that even with multiple shots, only the shot with the lowest $\tau$ plays a decisive role.

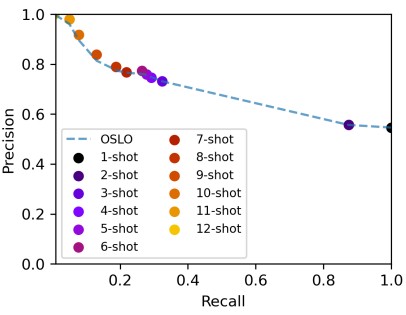

Figure 7: **Precision-Recall curve illustrating the attack performance of OSLO using more shots.**

## 6 Discussion

### 6.1 Why OSLO outperforms previous approaches

In this section, we discuss why previous attacks fail to achieve high precision or TPR under low FPR conditions, and how OSLO outperforms them. The boundary attack employs a global threshold of perturbation magnitude to differentiate between members and non-members. However, regardless of the chosen threshold, a considerable proportion of non-members are always found among the samples that require perturbations above this threshold (see the cumulative distribution function (CDF) in Figure 8a). On the other hand, by lowering the validation threshold $\tau$ in OSLO, the proportion of non-members among the failed adversarial attack samples significantly decreases, effectively squeezing out the non-members. For example, when classifying 4.4% of the samples as members, the

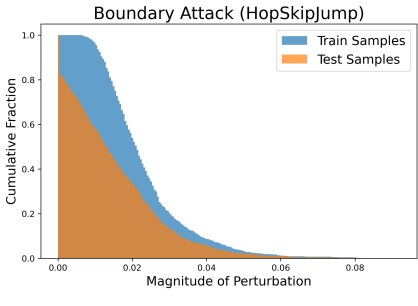

(a) Boundary Attack (HopSkipJump)

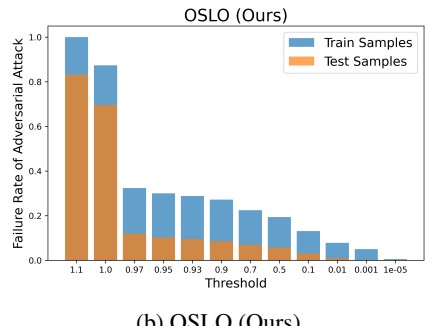

(b) OSLO (Ours)

Figure 8: **Comparison of boundary attack and OSLO regarding proportions of true positives (accurately identified members) and false positives (misclassified non-members) under varying threshold settings on CIFAR-10.**

boundary attack still includes 39% non-members (false positives) among those identified as members, whereas OSLO has only 10% non-members, resulting in 90% attack precision. We further discuss this by measuring the magnitude of the adversarial perturbation added to members and non-members in boundary attack versus OSLO on CIFAR-100, details are in Appendix A.2.

## 6.2 Mitigation

In addition to demonstrating the effectiveness of OSLO, we also evaluated six defensive strategies [41, 42, 43] aimed at mitigating membership privacy leakage. These experiments were conducted on the CIFAR-10 dataset using ResNet18 as the target model.

We trained three target models with different hyperparameters for each defense mechanism that requires retraining. Detailed configuration settings are provided in Appendix A.1.5. The results are presented in Figure 9. It can be observed that confidence alteration methods, such as MemGuard [41], are ineffective against OSLO, as expected. Other methods that mitigate memorization require strong regularization to be effective but result in a decrease in the

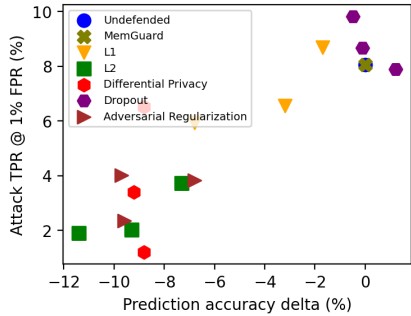

Figure 9: **Attack TPR under 1% FPR of OSLO against various defense mechanisms.**

model's test accuracy. Additionally, we tested OSLO against target models trained with adversarial training (details in Appendix A.5), and observed that while adversarial training reduces OSLO's effectiveness to some extent, it is not sufficient as a standalone defense.

## 7 Conclusions and limitations

In this paper, we propose OSLO, One-Shot Label-Only Membership Inference Attack (MIAs). OSLO is based on transfer-based adversarial attacks and identifies members by the different perturbations required when a sample is a member versus a non-member. We empirically demonstrate that OSLO not only operates under label-only settings with a single query but also significantly outperforms all previous label-only attacks by a large margin in terms of true positive rate (TPR) under the same false positive rate (FPR) and attack precision. Thus, OSLO sets a new benchmark for label-only MIAs and can serves as an effective measure of privacy leakage in these settings. We also delve into why OSLO outperforms existing methods. Furthermore, we evaluate multiple defenses against OSLO, highlighting its robustness.

OSLO requires training surrogate models that need an auxiliary dataset from the same distribution as the target model's training set. Although this is a common assumption in MIAs [1, 5] and can be alleviated by using synthetic data as demonstrated in prior work [1], it remains a limitation.

## Acknowledgments

The work was supported in part by the NSF grant 2131910, and by DARPA under Grant DARPA-RA-21-03-09-YFA9-FP-003.

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

# A  Appendix

## A.1  Additional Experimental Details

### A.1.1  Dataset Split

For CIFAR-10 and CIFAR-100, we used half of the training set (25,000 samples) to train the target model and the remaining half (25,000 samples) to train the source and validation models. The source and validation models were trained on the same data but remained disjoint from the data used to train the target model. Similarly, for SVHN, we created two disjoint subsets of 5,000 samples each, one for training the target model and the other for the surrogate models.

For the evaluation of the attacks, we randomly selected 1,000 samples from both the target model's training set and the remaining unused samples as target samples. The dataset split is summarized in Table 4.

Table 4: **Summary of dataset splits for training and evaluation.**

| Dataset | Target model training | Source/validation model training | Target samples |
|---|---|---|---|
| CIFAR-10 | 25,000 | 25,000 | 1,000 (members) + 1,000 (non-members) |
| CIFAR-100 | 25,000 | 25,000 | 1,000 (members) + 1,000 (non-members) |
| SVHN | 5,000 | 5,000 | 1,000 (members) + 1,000 (non-members) |

### A.1.2  Training setup

We consider ResNet18[13] and DenseNet121 [40] as target models, and additionally, Inception [44], VGG13[45], ResNeXt50[46], ShuffleNet [47], and PreActResNet18[48] are utilized as surrogate models. In our default setup, all models are trained for 100 epochs using the Adam optimizer, with a learning rate of $0.001$. We apply an $L2$ weight decay coefficient of $10^{-6}$ and use a batch size of 128.

All our models and methods are implemented in PyTorch. Our experiments are conducted on an NVIDIA GeForce RTX 2080 Ti with 20 GB of memory. The maximum training time for each surrogate or target model does not exceed 1.9 hours.

### A.1.3  OSLO Configuration.

The OSLO setup primarily includes the configuration of source and validation models, as well as the parameters for generating adversarial examples. By default, we use Inception [44], ResNeXt50 [46], and PreActResNet18 [48] as the source model architectures. For validation models, we use DenseNet121 [40], VGG13 [45], and ShuffleNetV2 [47] when ResNet18 [13] is the target model, and ResNet18, VGG13, and ShuffleNetV2 when DenseNet is the target model. We follow the assumption that the attacker has no knowledge of the target model architecture and does not include the same architecture in the source/validation models. Details on the default number of source and validation models are summarized in Table 5.

Table 5: **Source and validation model configurations per dataset in the default OSLO setup.**

| Dataset | Source models | Validation models | Total surrogate models |
|---|---|---|---|
| CIFAR-10 | Inception $\times$ 1
ResNeXt50 $\times$ 1
PreActResNet18 $\times$ 1 | DenseNet121 $\times$ 3
VGG13 $\times$ 3
ShuffleNetV2 $\times$ 3 | 12 |
| CIFAR-100 | Inception $\times$ 1
ResNeXt50 $\times$ 1
PreActResNet18 $\times$ 1 | DenseNet121 $\times$ 5
VGG13 $\times$ 5
ShuffleNetV2 $\times$ 5 | 18 |
| SVHN | Inception $\times$ 1
ResNeXt50 $\times$ 1
PreActResNet18 $\times$ 1 | DenseNet121 $\times$ 1
VGG13 $\times$ 1
ShuffleNetV2 $\times$ 1 | 6 |

More combinations of source and validation models are discussed in Appendix A.3. For generating adversarial examples, we set the maximum perturbation to $80/255$ and the number of sub-procedures to 80, increasing $\epsilon$ by $1/255$ each iteration.

### A.1.4 Baseline attack configuration

We summarize the configurations of the baseline attacks here. For the transfer attack, we use a relabeled dataset of the same size as the target model's training set and follow the original pipeline [6] to label the shadow dataset and train the shadow model. For boundary attacks based on adversarial perturbation (HopSkipJump and QEBA), we set a query budget of 6,000 per sample. For the Gaussian noise-based boundary attack, we set a query budget of 700. We use rotation and translation to construct augmented samples for the data augmentation attack, following the original method described in the paper [7].

### A.1.5 Defense Setup

This section outlines the hyperparameters used for the defenses evaluated against OSLO in Section 6.2. For each defense, three different hyperparameters were used to train the defended models, as detailed in Table 6.

Table 6: **Hyperparameter configurations for defenses evaluated against OSLO.**

| Defense | Hyperparameter | Values |
|---|---|---|
| L2 Regularization | Regularization parameter | {0.01, 0.005, 0.001} |
| L1 Regularization | Regularization parameter | {5e-6, 1e-5, 5e-5} |
| Adversarial Regularization | Alpha | {5, 6, 7} |
| Dropout | Dropout rate | {0.3, 0.5, 0.7} |
| DPSGD | Norm clipping bound | 1.2 |
| | Noise multiplier | {0.005, 0.01, 0.05} |

### A.2 Adversarial perturbation magnitude for members and non-members

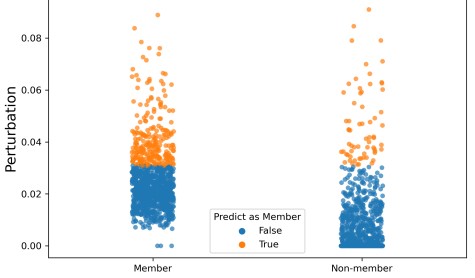
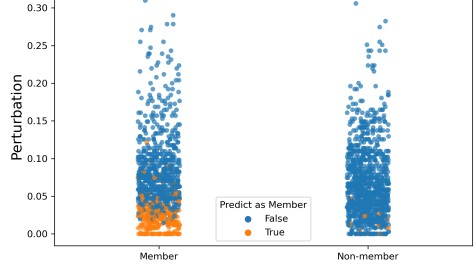

(a) Boundary Attack (HopSkipJump)  (b) OSLO (Ours)

Figure 10: **The magnitude of adversarial perturbations added to samples in the boundary attack and OSLO on CIFAR-100.** We adjusted the parameters of two attacks such that they both classify 15.8% of the samples as members (marked as yellow). The precision of OSLO is 95.2%, while the precision of boundary attack is 79.3%.

In Section 6.1, we discussed why OSLO outperforms. We further explore this by measuring the perturbation magnitude added to members versus non-members in both boundary attack and OSLO. The results are presented in Figure 10. We set $\tau = 0.01$ for OSLO, which resulted in classifying 15.8% of the samples as members. We also adjusted the parameters for the boundary attack to select the same proportion of samples classified as members.

From Figure 10a, it can be observed that using a single threshold to attempt to distinguish members from non-members leads to a significant proportion of non-members among the samples classified as members, regardless of the threshold value selected (i.e., drawing a horizontal line in the figure,

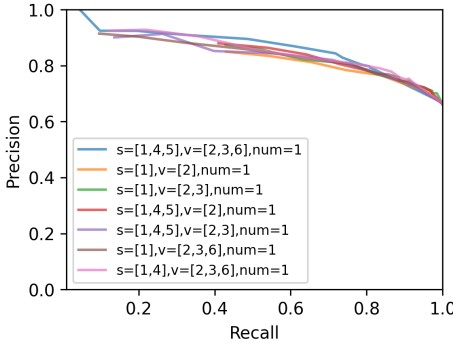
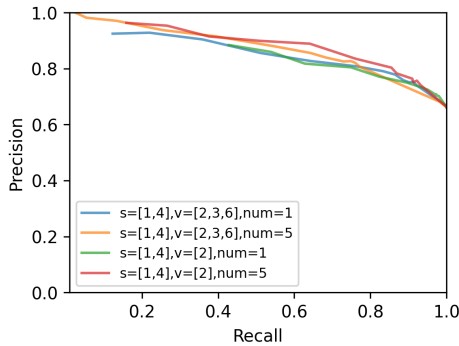

(a) Comparison of different source and validation model combinations

(b) Comparison of different numbers of models for each validation model architecture

Figure 11: **Precision-Recall curve comparing the attack precision and recall for OSLO across various combinations of source and validation models.**

with those above the line classified as members). For example, as shown in Figure 10a, 20.7% of the samples classified as members by the boundary attack are actually non-members. Conversely, OSLO conducts a hypothesis test for each sample, enabling it to accurately identify members even if their perturbations are relatively low compared to other samples. In Figure 10b, only 4.8% of the samples classified as members by OSLO are non-members.

## A.3 Effect of combinations of different source and validation models

We tested the effect of using different source model architectures and validation model structures on OSLO when ResNet18 [13] is the target model. Specifically, we considered the following model structures: DenseNet121 ("1") [40], Inception-V3 ("2") [44], VGG13 ("3") [45], ResNeXt50 ("4") [46], ShuffleNet-V2 ("5") [47], and Pre-Activation ResNet ("6") [48]. We selected different combinations from these models as source models or validation models. As shown in Figure 11a, different combinations have no significant impact on the trade-off between precision and recall, although they may affect the maximum achievable precision. For example, beyond a certain point, it becomes impossible to trade recall for additional precision. In such cases, we tested the effect of increasing the number of models (num) trained for each validation model architecture. As depicted in Figure 11b, increasing the number of validation models helps to make the trade-off curve more complete, as OSLO requires adding more perturbations to bypass more validation models. This helps to further reduce recall and increase precision.

## A.4 Further discussion on threshold $\tau$

### A.4.1 Using $\tau$ as the stopping criterion

In OSLO, the generation of adversarial perturbations stops when the confidence of the correct class on the validation model drops below a specified threshold, $\tau$. An intuitive alternative is to stop at the point of label flipping. However, we found that this approach does not guarantee sufficient perturbation to achieve a low FPR. To illustrate this, we compared different stopping criteria, including label flipping, across various thresholds. Table 7 shows the results for ResNet18 on CIFAR-10.

Table 7: **Comparison of different stopping criteria on OSLO's performance on CIFAR-10 using ResNet18.**

| Stopping criterion | TPR | FPR |
|---|---|---|
| Threshold $\tau = 0.1$ | 13.1% | 3.0% |
| Threshold $\tau = 0.01$ | 7.8% | 0.9% |
| Label flip | 61.0% | 30.8% |

As shown in Table 7, a small threshold $\tau$ is required to achieve a low FPR, whereas stopping at label flipping leads to a high FPR. The use of threshold $\tau$ provides better control over the TPR-FPR trade-off, which is crucial for OSLO's effectiveness.

### A.4.2 Choosing $\tau$ in realistic settings

In real-world scenarios, attackers can adapt $\tau$ based on the desired FPR. In OSLO, $\tau$ is calibrated by adjusting the success rate of the transfer-based attack on the target model. For instance, to achieve a 5% FPR, $\tau$ can be set to reach a 95% success rate. This approach may require access to a small set of non-members, which is a common assumption in MIAs [4, 30].

### A.5 Effectiveness of OSLO against models trained with adversarial training

Since OSLO relies on adversarial perturbations, increasing model robustness through adversarial training appears to be a natural defense strategy. To explore this, we evaluated OSLO against target models trained with adversarial training.

First, we tested OSLO's performance when the target models were trained with adversarial training, while the surrogate models were not. The results are presented in Table 8. We observed that adversarial training reduces OSLO's ability to achieve low FPR, but this comes at the cost of decreased model utility and increased training time.

Table 8: **OSLO's performance against target models trained with adversarial training on CIFAR-10.**

| Model | Threshold $\tau$ | Test ACC (%) | Training time (min) | Attack (%) | |
| --- | --- | --- | --- | --- | --- |
| | | | | TPR | FPR |
| ResNet18 (no adversarial training) | 0.01 | 82.6 | 23 | 7.8 | 0.9 |
| | 0.001 | | | 5.0 | 0.2 |
| ResNet18 (adversarial training, $\epsilon$=4/255) | 0.01 | 73.8 | 85 | 97.8 | 57.4 |
| | 0.001 | | | 95.6 | 52.7 |

We then evaluated OSLO's performance when both the target model and the surrogate models were trained with adversarial training. The results are shown in Table 9. OSLO remains effective, particularly in maintaining low FPR in this setting. These findings suggest that adversarial training alone may not be an adequate defense against OSLO.

Table 9: **Comparison of OSLO's performance when surrogate models are trained with and without adversarial training.**

| Target model training | Surrogate model training | Threshold $\tau$ | Attack TPR | Attack FPR |
| --- | --- | --- | --- | --- |
| Adversarial training ($\epsilon$=4/255) | No adversarial training | 0.001 | 95.6% | 52.7% |
| Adversarial training ($\epsilon$=4/255) | Adversarial training ($\epsilon$=4/255) | 0.001 | 15.7% | 0.7% |

### A.6 Comparison with YOQO

OSLO shares a similar goal with the recent work YOQO [34] in improving query efficiency for label-only MIAs. Below, we discuss the key differences to highlight our contributions.

**Effectiveness at low FPR.** It is widely accepted that MIAs should achieve low FPR to be practical [5, 11, 33]. However, by design, YOQO cannot adjust the TPR-FPR trade-off, making it unable to achieve low FPRs, as noted in their paper [34]. In contrast, OSLO is specifically designed to achieve

high TPRs under low FPRs and significantly outperforms existing methods in this metric. Thus, we are not able to have a fair comparison because YOQO cannot achieve the low FPR that OSLO primarily works on.

**Methodology.** To the best of our knowledge, OSLO is the first to leverage transfer-based adversarial attacks for label-only MIAs. YOQO, on the other hand, shares a similar approach with LiRA [5], relying on shadow models trained on target samples (IN models) and shadow models not trained on the target samples (OUT models). This approach depends on the similarity between the shadow model and target model architectures and is sensitive to architecture differences [34]. In contrast, OSLO is robust to variations in surrogate model architecture, as demonstrated by our experiments.

