# OpenReview forum: "OSLO: One-Shot Label-Only Membership Inference Attacks"
_NeurIPS.cc/2024/Conference — NeurIPS 2024 poster_

### Official Review · Reviewer_CtKQ · 2024-06-26

**Soundness:** 2
**Presentation:** 3
**Contribution:** 3
**Rating:** 5
**Confidence:** 3

**Summary:**

In this work, the authors propose a novel label-only membership inference attack. This method involves adding an adversarial perturbation to compel a model, trained without the target data point, to misclassify the image. This approach requires only a single query to the model and achieves SOTA performance for label-only attacks.

**Strengths:**

- The paper is well-written, with a strong and compelling motivation.
- The results are very promising.

**Weaknesses:**

The main concern for me is that this paper lacks important related works, like [1] and [2]. In particular, [1] uses a similar strategy to conduct a one-shot label-only inference attack. While the method proposed in this paper is quite similar to [1], it seems to improve the transferability of the adversarial perturbation by checking holdout models. However, the overall contribution is not clear to me. I am not sure if the contribution is significant enough for NeurIPS, but I would appreciate hearing the authors' perspectives on this.

[1] WU, YUTONG et al. You Only Query Once: An Efficient Label-Only Membership Inference Attack. In The Twelfth International Conference on Learning Representations. https://openreview.net/forum?id=7WsivwyHrS&referrer=%5Bthe%20profile%20of%20Jiwei%20Li%5D(%2Fprofile%3Fid%3D~Jiwei_Li1)

[2] Wen, Yuxin et al. Canary in a Coalmine: Better Membership Inference with Ensembled Adversarial Queries. ArXiv abs/2210.10750 (2022): n. pag.

**Questions:**

- What if the target model has a different training algorithm, like SGD? Would this make the attack less effective?
- How to apply this attack to a new target data point? Does the attacker have to retrain some of the models?

**Limitations:**

The authors mention limitations at the end of the paper which I appreciate.

---

> ### Author Rebuttal · Authors · 2024-08-07
>
> ---
> **To Reviewer CtKQ:**
>
> **Response to Weaknesses 1:**
> Thank you for your feedback. We apologize for not including a comparison with YOQO [1] in our initial submission. Please note that OSLO and YOQO differ in several key aspects:
>
> - **Effectiveness on Low FPR Settings:** The accepted norm in the MIA community is that achieving low FPR is key for MIAs to be practical [2][3][4]. By design, YOQO cannot adjust the TPR and FPR trade-off and thus **cannot achieve low FPRs**, which is also acknowledged in their paper [1]. We reproduced YOQO's results on ResNet18 with 2,500 training samples (as used in their paper), and although it achieved over 80% attack accuracy, its FPR was 26%, which makes it highly impractical. On the other hand, OSLO is designed to achieve high attack TPR under low FPR conditions and significantly outperforms previous methods on this metric. Thus, we are not able to have a fair comparison because YOQO cannot achieve the low FPR that OSLO primarily works on.
>
> - **Methodology:** To our knowledge, OSLO is the first to use transfer-based adversarial attacks to create label-only attack samples. In contrast, YOQO is a variant of the previous work LIRA [3] in a label-only setting, requiring the training of models on the target sample (IN models) and models not trained on the target sample (OUT models). It optimizes the differences between IN and OUT models to construct attack samples.
>
> - **Assumptions about the Attacker:** OSLO does not assume that the attacker knows the target model's architecture. In our experiments, the surrogate model used has a different architecture from the target model. YOQO, however, relies on the similarity between shadow models and the target model, and the difference in architecture significantly impacts its effectiveness [1].
>
> We will include these comparisons in the revised version of our paper.
>
> Regarding the comparison with [2], [2] and OSLO operate in different settings. [2] requires access to the target model's logits, whereas OSLO operates in a label-only context. In this paper, we only compare with label-only MIAs, but we will include a discussion of [2] in the related work section. We hope this clarifies our contributions and the significance of our work.
>
> **Response to Question 1:**
> Thank you for your question. We tested the effectiveness of OSLO against target models trained with different algorithms and found that OSLO remains highly effective. Below are the results for CIFAR-10 using ResNet18, where the target model was trained with SGD, and the surrogate models were trained with Adam:
>
> | Target Model | Source and Validation Models | Attack Precision (Recall > 1%) | Attack TPR @ 1% FPR |
> |-----------------------------|------------------------------------------------|-------------------------------|--------------------|
> | SGD, lr=0.01                     | Adam, lr=0.001                                 | 95.9%                         | 10.0%              |
> | SGD, lr=0.1                      | Adam, lr=0.001                                 | 95.3%                         | 9.2%               |
>
> These results demonstrate that **OSLO is effective even when the target model is trained using a different algorithm**. We will include these findings in the revised version of the paper.
>
> **Response to Question 2:**
> Thank you for your question. OSLO does not require retraining any models when applied to a new target data point. For a given target model, OSLO trains the source models and validation models **only once**. These models can then be used to add perturbations to all target samples. For a new target data point, we compute gradient updates on the source models and validate using the pre-trained validation models. This process uses the already trained models, so no retraining is needed.
>
> **References:**
>
> [1] Wu et al. You Only Query Once: An Efficient Label-Only Membership Inference Attack. ICLR’24
> [2] Wen et al. Canary in a Coalmine: Better Membership Inference with Ensembled Adversarial Queries. ICLR’23
> [3] Carlini et al. Membership Inference from First Principles. S&P’22
> [4] Chaudhari et al. Chameleon: Increasing Label-Only Membership Leakage with Adaptive Poisoning. ICLR’24

---

> > ### Comment · Reviewer_CtKQ · 2024-08-08
> > **Thank you**
> >
> > Thank the authors for the clarification. I do see the distinction between [1, 2] and the novelty of the paper now. I really appreciate it. Therefore, I increase my score accordingly.
> >
> > Meanwhile, I have one question: how do you control the FPR accurately? I see you can control the threshold $\tau$ to change FPR, but how do you get the exact value, such as FPR = 0.1%?

---

> > > ### Author Response · Authors · 2024-08-09
> > > **Thank you for your feedback**
> > >
> > > Thank you for your positive feedback and for raising this insightful question.
> > >
> > > To address your question, we controlled the FPR by selecting multiple threshold values (tau) to plot the complete ROC curve. Our approach, OSLO, is capable of achieving an FPR of less than 0.1%, and in most cases, even a 0% FPR. We then identify and report the TPR corresponding to an FPR of 0.1% on the curve. This process is analogous to adjusting the loss to generate a complete ROC curve for loss-based attacks. We hope this explanation clarifies the method we employed. Please let us know if you have any further questions.

---

### Official Review · Reviewer_Vvp4 · 2024-06-26

**Soundness:** 3
**Presentation:** 3
**Contribution:** 3
**Rating:** 7
**Confidence:** 5

**Summary:**

The paper proposes a new label-only membership inference attack that only needs a single query to the target model to perform the attack. The general idea of the proposed attack is based on the observation that member samples need more adversarial perturbation to force a misclassification than non-member samples.
The adversarial perturbation leads to non-members being misclassified while members are more robust and are still correctly classified. Transferable adversarial attacks are used to perform the attack in a single query using several surrogate models. The proposed One-Shot Label Only (OSLO) attack has three phases: 1) surrogate model training 2) creating the transferable adversarial examples on the surrogate models and 3) performing the membership inference on the target model. In contrast to other label-only membership inference attacks, OSLO has a different perturbation budget for each sample.
The proposed attack is evaluated on three datasets (CIFAR-10, CIFAR-100, and SVHN), and in an ablation study, it is shown that the attack is model agnostic and also works with other transferable adversarial attack methods. The paper is technically sound, and the results show that OSLO outperforms existing attacks by a large margin.

**Strengths:**

- The paper is well-written and easy to understand
- The paper is well organized, and even a reader who is not an expert in privacy attacks can follow the paper easily
- The proposed approach is novel
- The approach is quite simple yet apparently highly effective

**Weaknesses:**

- A recent paper also proposes a label-only attack that needs only a single query [1]. However, OSLO is not mentioned and, more importantly, not compared to YOQO.
- Equation 2 is presented without introducing any of the variables. It is easier to understand equation 2 if it is mentioned in the text what $h_y(x;\theta)$ and $h_j(x;\theta)$ is.

[1] You Only Query Once: An Efficient Label-Only Membership Inference Attack, Wu et al., ICLR 2024

Misc:
- I think the H_0 and H_1 hypotheses have to be switched in equation 1
- There seems to be a typo in the heading of 6.1. I think it should be "Why OSLO outperform[s] previous approaches".

**Questions:**

*Q1:* Why is the generation of the adversarial example stopped when the confidence is below a certain threshold $\tau$? In my understanding, it would make more sense to stop the generation as soon as the label flips.
*Q2:* In reality, the attacker has to choose the threshold $\tau$. How would the attacker choose the threshold in a realistic setting?
*Q3:* The perturbation budget is chosen differently for each sample. On the other hand, the threshold $\tau$ is the same for all the samples. Wouldn't it make sense to choose a different threshold for different samples since a lower threshold means more perturbation is added to the samples?
*Q4:* In 6.2, it is mentioned that "three target models using different hyper-parameters" were trained. However, the hyperparameters were never mentioned. Which model architectures were trained here, what were the hyper-parameters for training these models, and what parameters were altered?

**Limitations:**

The limitations are addressed.

---

> ### Author Rebuttal · Authors · 2024-08-07
>
> ---
> **To Reviewer Vvp4:**
>
> **Response to Weaknesses 1:**
> Thank you for your insightful feedback. We apologize for not including a comparison with YOQO in our paper. Please note that the accepted norm in the MIA community is that achieving low FPR is key for MIAs to be practical [1][2]. Although YOQO also requires only a single query, it is designed in a way that it cannot adjust the TPR and FPR trade-off, and thus **cannot achieve low FPRs** (as acknowledged in their paper [3]), making YOQO highly impractical. For example, we reproduced YOQO's experiments on ResNet18 using 2,500 training samples (as per their settings), and although the attack accuracy was over 80%, the FPR was 26%. In contrast, OSLO is primarily designed to achieve high TPR under low FPR conditions. Therefore, we are not able to have a fair comparison because YOQO cannot achieve the low FPRs that OSLO achieves.
>
> **Response to Weaknesses 2/3/4:**
> Thank you for pointing these out. We will clarify and fix these issues in our paper, including the introduction of variables in Equation 2 and correcting the typos.
>
> **Response to Question 1:**
> Thank you for your suggestion. In our experiments, we found that stopping the generation of the adversarial example as soon as the label flips does not ensure that enough perturbation is added to achieve a sufficiently low FPR. For example, with ResNet18:
>
> | Threshold $\tau$ | TPR  | FPR  |
> |------------------|------|------|
> | 0.1              | 13.1%| 3.0% |
> | 0.01             | 7.8% | 0.9% |
> | Label Flip       | 61.0%| 30.8%|
>
> We observed that a small $\tau$ is needed to achieve a low FPR. We will include experimental comparisons regarding this in the final version of the paper. Thank you for bringing this up.
>
> **Response to Question 2:**
> Thank you for your question. We believe the attacker can choose $\tau$ based on the desired FPR. By adjusting the success rate of the transfer-based adversarial attack on the target model, the attacker can set $\tau$ accordingly. For example, if the attacker tolerates an FPR of 5%, they can adjust $\tau$ to achieve a 95% success rate on the target model. This may introduce an additional assumption by requiring a small set of non-members to detect the success rate of the adversarial attack; however, access to some non-members is a common assumption in MIAs [4][5]. We will include this discussion in the next version of the paper.
>
> **Response to Question 3:**
> We agree that dynamic adjustment of the threshold is a promising direction for improving OSLO. However, it is challenging to determine how to adjust the threshold without prior knowledge of the target sample's membership. The high-level idea of OSLO is to choose a sufficiently small $\tau$ to ensure all non-members are misclassified, thus identifying members who are correctly classified. Currently, we use the same threshold for all samples based on this intuition.
>
> **Response to Question 4:**
> Thank you for pointing this out. In our experiments, we used ResNet18 on CIFAR-10. The following hyperparameters were varied:
>
> | Defense Method                    | Hyperparameter                          | Values                              |
> |-----------------------------------|-----------------------------------------|-------------------------------------|
> | L2 Regularization       | Regularization parameter                | {0.01, 0.005, 0.001}                |
> | L1 Regularization       | Regularization parameter                | {5e-6, 1e-5, 5e-5}                  |
> | Adversarial Regularization | Alpha                                  | {5, 6, 7}                           |
> | Dropout                           | Dropout rate                            | {0.3, 0.5, 0.7}                     |
> | DPSGD| Norm clipping bound                     | 1.2                                 |
> |                                   | Noise multiplier                        | {0.005, 0.01, 0.05}                 |
>
>
> We will further clarify the model structures and hyperparameters used in the defenses mentioned in Section 6.2 of the revised paper.
>
> **References:**
>
> [1] Carlini et al. Membership Inference from First Principles. S&P’22
> [2] Chaudhari et al. Chameleon: Increasing Label-Only Membership Leakage with Adaptive Poisoning. ICLR’24
> [3] Wu et al. You Only Query Once: An Efficient Label-Only Membership Inference Attack. ICLR’24
> [4] Salem et al. ML-Leaks: Model and Data Independent Membership Inference Attacks and Defenses on Machine Learning Models. NDSS’19
> [5] Hui et al. Practical Blind Membership Inference Attack via Differential Comparisons. NDSS’21

---

> > ### Comment · Reviewer_Vvp4 · 2024-08-11
> > **Response to Rebuttal**
> >
> > Thank you very much for your detailed rebuttal and the additional insights!
> >
> > My questions have been appropriately addressed and I will retain my current score.

---

> > > ### Author Response · Authors · 2024-08-11
> > > **Thanks for your response**
> > >
> > > Thank you for your thorough review and for taking the time to consider our responses. We appreciate your acknowledgment of our work. Your feedback has been invaluable in improving the clarity and quality of our paper.

---

### Official Review · Reviewer_B8Vg · 2024-07-12

**Soundness:** 2
**Presentation:** 3
**Contribution:** 2
**Rating:** 6
**Confidence:** 2

**Summary:**

The paper investigates the vulnerability of deep learning models to membership inference attacks (MIAs), focusing on label-only settings where only the predicted hard label is available. It introduces the One-Shot Label-Only (OSLO) MIA, which infers a sample's membership in the training set with a single query. OSLO leverages transfer-based black-box adversarial attacks, exploiting the characteristic that member samples are more resistant to adversarial perturbations. The paper presents extensive comparisons against state-of-the-art label-only attacks, showing that OSLO significantly outperforms them in terms of precision and true positive rate (TPR) under low false positive rates (FPR). For instance, OSLO achieves a TPR that is 7x to 28x stronger than previous methods under a 0.1% FPR on CIFAR10 for a ResNet model, while maintaining over 95% precision across multiple datasets. Additionally, the paper evaluates various defenses against OSLO, demonstrating its robustness and effectiveness.

**Strengths:**

- The research question is important and well-motivated.

- The paper is clearly written and easy to understand.

- The proposed OSLO method significantly outperforms previous approaches.

**Weaknesses:**

- **Mitigation Strategies**: The discussion on potential countermeasures against OSLO is somewhat limited. A deeper exploration of feasible strategies to mitigate the risks posed by such attacks would enhance the paper's practical utility.

- **Computational Costs**: While OSLO does not rely on uniform perturbations like some existing methods, it requires higher computational resources to calculate specific perturbations for each sample. This could be a significant drawback in scenarios with limited computational capacity.

**Questions:**

- How would methods aimed at enhancing model robustness, such as adversarial training, impact the performance of label-only inference attacks?

**Limitations:**

The authors have adequately summarized the limitations at the end of the paper.

---

> ### Author Rebuttal · Authors · 2024-08-07
>
> ---
> **To Reviewer B8Vg:**
>
> **Response to Weakness 1:**
> Thank you for your suggestion. We agree that deeper research into defenses against such attacks is a promising direction. However, due to space limitations, we focused our contribution on improving label-only MIAs in terms of query cost and attack performance. We will leave further exploration of defenses for future work.
>
> **Response to Weakness 2:**
> Thank you for your insightful feedback. The calculation of perturbations in OSLO occurs locally, and the time required for each sample is limited (less than 3 minutes on CIFAR-10). We believe this computational cost is acceptable for an attack setting. Once the calculations are complete, only one query to the target model is needed by the attacker, making the attack highly practical against target models under almost any resource constraints.
>
> It is important to note that previous methods, while using uniform thresholds, still require calculating specific perturbations for each sample. This process involves continuous access to the target model, resulting in potentially thousands of queries [1][2]. This can be impractical for resource-limited target models. In contrast, OSLO minimizes the number of queries to the target model, thus offering a more efficient approach.
>
> **Response to Question 1:**
> Thank you for your question. We tested OSLO against target models trained with adversarial training. The results are as shown in the table below:
>
> | Model                                    | Threshold $\tau$ | Test ACC | Training time (min) | Attack TPR | Attack FPR |
> |------------------------------------------|-----------|----------|---------------------|------------|------------|
> | ResNet18 (no adversarial training)       | 0.01      | 82.6%    | 23                  | 7.8%       | 0.9%       |
> |                                          | 0.001     |          |                     | 5%         | 0.2%       |
> | ResNet18 (Adversarial training, ε=4/255) | 0.01      | 73.8%    | 85                  | 97.8%      | 57.4%      |
> |                                          | 0.001     |          |                     | 95.6%      | 52.7%      |
>
> We observed that adversarial training can reduce OSLO's ability to achieve low FPR, but it **also reduces model utility and increases training overhead**. Additionally, it increases the overall difference in robustness to adversarial examples between members and non-members, potentially leading to improved attack effectiveness on average metrics such as attack accuracy. We will discuss these aspects further in the next version of the paper.
>
> **References:**
>
> [1] Choquette-Choo et al. Label-Only Membership Inference Attacks. ICML’21.
> [2] Li et al. Membership leakage in label-only exposures. CCS’21

---

> > ### Comment · Reviewer_B8Vg · 2024-08-10
> > **Thanks for the response**
> >
> > Thank you for the response. The findings on the influence of adversarial training are indeed intriguing, and I appreciate your work on this.
> >
> > I have one additional question: when the victim model has undergone adversarial training, should the surrogate models used by the attacker also be adversarially trained? If so, does this imply that the attacker must first determine whether the victim model is adversarially trained before mounting the MIA? I understand that the inclusion of adversarial training complicates the scenario, as the adversarial examples used during training could also be considered member instances.

---

> > > ### Author Response · Authors · 2024-08-10
> > > **Thank you for your follow-up question**
> > >
> > > Thank you for your insightful question. In our previous response to Question 1, we presented results where only the victim model was adversarially trained while the surrogate models were not. To address your question more thoroughly, we conducted additional experiments where both the victim model and the surrogate models were adversarially trained:
> > >
> > > | **Victim Model Training**                | **Surrogate Model Training**              | **Threshold (tau)** | **TPR** | **FPR** |
> > > |------------------------------------------|-------------------------------------------|---------------------|---------|---------|
> > > | Adversarial Training (ε=4/255)           | No Adversarial Training                   | 0.001               | 95.6%   | 52.7%   |
> > > | Adversarial Training (ε=4/255)           | Adversarial Training (ε=4/255)            | 0.001               | 15.7%   | 0.7%    |
> > >
> > > These findings suggest that OSLO remains effective, particularly in maintaining low FPR, when both the victim model and the surrogate model are adversarially trained. However, as you mentioned, this may require the attacker to first determine whether the victim model has been adversarially trained. In contrast, when the surrogate model is not adversarially trained, the overall gap in robustness is larger, potentially leading to better performance on average metrics such as attack accuracy. Thus, depending on the attacker's knowledge and assumptions, different strategies can be employed, leading to different outcomes.
> > >
> > > We will include these findings in the revised paper. Thank you again for your valuable feedback. Please feel free to reach out if you have any further questions.

---

> > > > ### Comment · Reviewer_B8Vg · 2024-08-11
> > > > **Thank you**
> > > >
> > > > Thank you for the prompt reply and for providing additional results with new insights. My concerns have been fully addressed, and I choose to keep my score.

---

> > > > > ### Author Response · Authors · 2024-08-11
> > > > > **Thanks for the positive feedback**
> > > > >
> > > > > Thank you for your continued engagement with our work. We’re glad that our responses have fully addressed your concerns. We will include the results provided during the rebuttal in the final version.

---

### Official Review · Reviewer_uq6v · 2024-07-12

**Soundness:** 3
**Presentation:** 3
**Contribution:** 2
**Rating:** 5
**Confidence:** 4

**Summary:**

This paper presents a new label-only membership inference attack, which requires only a single query to the target model. Specifically, it leverages transfer-based black-box attacks to generate an adversarial perturbation. This perturbation is then added to the sample and input into the target model, and its prediction is used to judge whether it is a member or non-member. The experiments demonstrate the effectiveness of the proposed methods.

**Strengths:**

- The comparison with previous label-only attacks is clear, making it easy to pinpoint the main contributions of this paper.

- The experiments are comprehensive, covering many different datasets and models.

- It is nice to see that the paper evaluates many defense strategies and uses Figure 9 to clearly show the ineffective handling of the model accuracy-attack mitigation trade-off in existing defenses.

**Weaknesses:**

- While the paper is motivated by improving the attack costs compared to existing works, this method requires training multiple source models and validation models, which could be quite costly. This is especially true when applying it to relatively large-scale models, such as diffusion models, which makes the proposed attacks less practical.

- It is unclear whether the proposed MIA was tuned on the validation data independent of the test set. Given that this method requires tuning threshold $\tau$, there is no mention of how this tuning is done and whether using the validation data. If test data was used to tune the method instead of validation data, the results reported might be overly optimistic and not reflect the proposed MIA's performance on "true" test (held-out) data.

- The experimental setting is unclear. Specifically, it remains uncertain how the data split is implemented. Given that 25,000 samples are used to train the target model, it is important to clarify how many datasets are used to train the source models and validation models. More importantly, how many source models and validation models are trained for the experiments? As this is closely tied to the attack costs.

- It is better to conduct an ablation study. The two main contributions are using transfer-based attacks to improve the query costs and using the validation models to adjust the scale of perturbations added to the adversarial example. The latter design is interesting as it resembles calibration to the sample difficulty in existing membership inference literature. To make each contribution clearer, it is better to conduct an ablation study to investigate the importance of each design. Additionally, an ablation study should investigate the impacts of the number of source models and validation models on the attack performance.

**Questions:**

- How does your proposed approach perform when tuning the method's components on validation data that is disjoint from the test data on which final performance is reported?

- How many datasets are used to train the source models and validation models?

- How many source models and validation models are trained for the experiments?

**Limitations:**

Please see my above comments.

---

> ### Author Rebuttal · Authors · 2024-08-07
>
> ---
> **To Reviewer uq6v:**
>
> **Response to Weaknesses 1:**
> Thank you for your feedback. While we acknowledge that training these surrogate models introduces additional computational overhead, it is a **one-time process** for each target model. Once completed, these models can be used to infer any target sample's membership, unlike previous online attacks [2][3] that require retraining shadow models for each sample.
>
> Moreover, training shadow or surrogate models is a standard practice in MIAs, including well-cited state-of-the-art works [1][2]. It is worth noting that OSLO requires fewer models compared to the related works. For instance, the state-of-the-art score-based MIA LiRA [2] requires training at least 128 shadow models, and the most recent label-only attack YOQO [3] requires training 32 shadow models. However, OSLO needs fewer than 20 surrogate models (e.g., 12 for CIFAR-10, 18 for CIFAR-100, and 6 for SVHN). To our knowledge, OSLO is the first label-only attack to achieve high TPR under low FPR. Therefore, we believe the computational cost is acceptable given the improvements.
>
> **Response to Weaknesses 2 / Question 1:**
> Instead of tuning our method on the test or validation data, we adjust $\tau$ to obtain and report the complete ROC curve (e.g., in Figures 2 and 3, each point on OSLO's ROC curve corresponds to a different $\tau$). The primary goal of this adjustment is to capture the trade-off between TPR and FPR rather than setting $\tau$ to a fixed value derived from any specific data set. This approach is analogous to adjusting loss thresholds to generate the ROC curve, as discussed in previous loss-based MIAs [2]. We are happy to provide further details during the interactive discussion if our explanation does not fully address your question.
>
> **Response to Weaknesses 3 / Question 2:**
> Thank you for pointing this out. We apologize for omitting the training details due to page limitations. For CIFAR-10 and CIFAR-100, we use half of the training set (25,000 samples) to train the target model and the remaining half (25,000 samples) to train the source models and validation models. The source models and validation models are trained on the same data but are disjoint from the data used to train the target model. Similarly, for SVHN, we use two disjoint subsets of 5,000 samples each to train the target model and the surrogate models. For the evaluation of the attack, we randomly select 1,000 samples from both the target model's training set and the remaining samples (not used for any model training) as target samples. Here is a summary table to clarify the data split:
>
> | Dataset    | Target Model Training Samples | Source/Validation Model Training Samples | Target Samples                                                 |
> |------------|-------------------------------|------------------------------------------|----------------------------------------------------------------|
> | CIFAR-10   | 25,000                        | 25,000                                   | 1,000 (from target model's training set) + 1,000 (from remaining samples) |
> | CIFAR-100  | 25,000                        | 25,000                                   | 1,000 (from target model's training set) + 1,000 (from remaining samples) |
> | SVHN       | 5,000                         | 5,000                                    | 1,000 (from target model's training set) + 1,000 (from remaining samples) |
>
> We will clarify these details and add them to the appendix in the revised manuscript.
>
> **Response to Weaknesses 4:**
> Thank you for your valuable suggestions. We appreciate your understanding of the innovative aspects of OSLO. Regarding the impact of the number of source models and validation models on the attack performance, We have found that increasing the number of source and validation models does not significantly improve the trade-off between TPR and FPR, or the trade-off between precision and recall, but it allows for further trading TPR to achieve lower FPR. We have included these discussions in Appendix A.3.
>
> Regarding the ablation study on the role of validation models, we are running experiments on that and will post the results once they become available during the interactive discussion.
>
> **Response to Question 3:**
> Thank you for your question. In our default settings, we train the following number of source and validation models:
> - CIFAR-10: 3 source models and 9 validation models (3 models per architecture)
> - CIFAR-100: 3 source models and 15 validation models (5 models per architecture)
> - SVHN: 3 source models and 3 validation models (1 model per architecture)
>
> We will provide a detailed description of the source and validation model configurations in the revised paper.
>
> **References:**
>
> [1] Shokri et al. Membership Inference Attacks against Machine Learning Models. S&P’17
> [2] Carlini et al. Membership Inference from First Principles. S&P’22
> [3] Wu et al. You Only Query Once: An Efficient Label-Only Membership Inference Attack. ICLR’24

---

> > ### Comment · Reviewer_uq6v · 2024-08-10
> >
> > Thank you for the detailed response. Regarding the adjustment of $\tau$ to obtain and report the complete ROC curve, I am curious whether this was done on the shadow or target datasets?

---

> > > ### Author Response · Authors · 2024-08-11
> > >
> > > Thank you for your follow-up. Below is our further clarification and results:
> > >
> > > **Clarification on adjusting $\tau$:** The adjustment of $\tau$ to generate the ROC curve was specifically intended to reflect the attack's performance on the target dataset, representing the "true" test or held-out data. We varied $\tau$ systematically to explore how the attack behaves on the target data under different $\tau$ settings. It’s important to note that this process did not involve directly using any validation or test set data for tuning $\tau$. The shadow dataset was only used to train the surrogate model, and once trained, it does not participate in the subsequent attack process.
> > >
> > > **Ablation study results:** As promised in our previous response to Weakness 4, we conducted the ablation study as you suggested. We tested OSLO on CIFAR-10 with ResNet18 without the validation model, where all samples used the same perturbation budget. The results are as follows:
> > >
> > > | Perturbation Budget Epsilon $\epsilon$ | TPR   | FPR   |
> > > |----------------------------------------|-------|-------|
> > > | **8/255**                              | 20.1% | 13%   |
> > > | **16/255**                             | 2.3%  | 1.4%  |
> > > | **32/255**                             | 0.1%  | 0.1%  |
> > >
> > > These results indicate that without the validation model, using a uniform perturbation budget $\epsilon$ for all samples leads to significantly reduced attack effectiveness. For example, at $\epsilon = 32/255$, nearly all members and non-members were not successfully classified, resulting in both TPR and FPR being very low. This highlights the critical role of validation models in effectively calibrating perturbations.
> > >
> > > We hope these clarifications and results address your concerns. Please let us know if you have any further concerns we can address.

---

> > > > ### Comment · Reviewer_uq6v · 2024-08-12
> > > >
> > > > Thank you for the additional details provided. Please consider including all the experiments from the rebuttal in the revised manuscript. I will raise my scores accordingly.

---

> > > > > ### Author Response · Authors · 2024-08-12
> > > > >
> > > > > Thank you for your encouraging feedback.  While NeurIPS guidelines do not permit us to update the paper at this time, we will make sure to incorporate all the experiments from the rebuttal in the final version. Your input has been invaluable in strengthening our work.

---

### Author Rebuttal · Authors · 2024-08-07

We thank the reviewers for their insightful feedback and constructive suggestions, which have helped us enhance the clarity and robustness of our paper. We are committed to addressing all concerns and providing a comprehensive and well-documented study in the revised version. Our responses to each of the reviewers' questions are detailed below.

---

### Decision · Program_Chairs · 2024-09-25

**Decision:**

Accept (poster)

**Comment:**

This paper proposes a new membership inference attack called OSLO, inspired by the empirical phenomenon of black-box adversarial attack transfer. Reviewers appreciated the clarity of writing and the empirical results, and the authors engaged with reviewers extensively during the rebuttal period in order to address most of the relevant concerns. I would strongly encourage the authors to include the additional experiments in the paper and additionally implement other suggested edits.